# Does a learner-centered approach using teleconference improve medical students' psychological safety and self-explanation in clinical reasoning conferences? a crossover study

Yoji Hoshina◉⊚, Kiyoshi Shikino◉*⊚, Yosuke Yamauchi‡, Yasutaka Yanagita‡, Daiki Yokokawa◉‡, Tomoko Tsukamoto‡, Kazutaka Noda‡, Takanori Uehara◉‡, Masatomi Ikusaka‡

Department of General Medicine, Chiba University Graduate School of Medicine, Chiba, Japan

⊚ These authors contributed equally to this work.
‡ YY, YY, DY, TT, KN, TU and MI also contributed equally to this work.
* kshikino@gmail.com

## Abstract

During clinical reasoning case conferences, a learner-centered approach using teleconferencing can create a psychologically safe environment and help learners speak up. This study aims to measure the psychological safety of students who are supposed to self-explain their clinical reasoning to conference participants. This crossover study compared the effects of two clinical reasoning case conference methods on medical students' psychological safety. The study population comprised 4th-5th year medical students participating in a two-week general medicine clinical clerkship rotation, from September 2019 to February 2020. They participated in both a learner-centered approach teleconference and a traditional, live-style conference. Teleconferences were conducted in a separate room, with only a group of students and one facilitator. Participants in group 1 received a learner-centered teleconference in the first week and a traditional, live-style conference in the second week. Participants assigned to group 2 received a traditional, live-style conference in the first week and a learner-centered approach teleconference in the second week. After each conference, Edmondson's Psychological Safety Scale was used to assess the students' psychological safety. We also counted the number of students who self-explained their clinical reasoning processes during each conference. Of the 38 students, 34 completed the study. Six out of the seven psychological safety items were significantly higher in the learner-centered approach teleconferences (p<0.01). Twenty-nine (85.3%) students performed self-explanation in the teleconference compared to ten (29.4%) in the live conference (p<0.01). A learner-centered approach teleconference could improve psychological safety in novice learners and increase the frequency of their self-explanation, helping educators better assess their understanding. Based on these results, a learner-centered teleconference approach has the potential to be a method for teaching clinical reasoning to medical students.

**Data Availability Statement:** All relevant data are within the paper and its S1, S2 Tables and S1–S4 Figs.

**Funding:** The authors received no specific funding for this work.

**Competing interests:** The authors have declared that no competing interests exist.

## Introduction

Clinical reasoning is a core competency for all medical experts, and it is crucial that all medical students learn this skill [1, 2]. However, inadequate knowledge, flaws in data gathering, and an improper approach to information processing make it a challenge for most novice learners [1]. The Department of General Medicine, Chiba University, Japan, traditionally holds weekly case conferences, using the serial-cue approach [3] (traditional, live style conference) to allow medical students to understand the clinical reasoning process of experts, and reach a diagnosis. When students ask or answer questions, they are provided opportunities for self-explanation [4, 5] regarding the case topic, so educators can assess their clinical reasoning process and provide appropriate real-time feedback that helps students reorganize knowledge [6].

However, most novices had difficulty speaking up in front of department staff during the conference, possibly because of the existence of a professional hierarchy in medicine, and the influence of Japanese culture [7], making students feel unsafe. Psychological safety, the degree to which people view the environment as conducive to interpersonally risky behaviors, such as speaking up or asking for help, impacts the degree to which individual or group learning can occur [8]. In fact, a profession-derived status hierarchy has a positive relationship with psychological safety [9, 10], and individuals who experience greater psychological safety are more likely to speak up and present new ideas [8, 10]. The degree of team psychological safety, which is a shared belief that the team is safe for interpersonal risk-taking, is key to facilitating learning behavior by alleviating excessive concern about others' reactions to actions that have the potential for embarrassment or threat [11]. Our goal was to create a clinical reasoning conference environment in which learners could maintain their psychological safety and comfortably share their ideas, so that educators could provide effective feedback.

Learner-centered approaches such as problem-based learning (PBL), where the emphasis is on students [12, 13] and which focus on small groups with a facilitator involving the learner as an active participant and encouraging the development of an in-depth approach to learning [14] can develop a group learning environment [15]. We assumed that such an environment would improve learners' psychological safety and allow them to speak up easily.

Teleconferences, which describe the creation of two or more learning environments where users can engage in real-time communication to exchange presentations or data via audio or live video [16], can constitute a learner-centered model by helping students learn on their own, as well as in collaboration with others [17]. A previous study has shown that teleconferences can support learners' autonomy, allowing trainees to engage more actively in the conference and creating a cooperative relationship with their trainers and co-trainees [17]. Although we could not find any previous studies that measured psychological safety during teleconferences, we hypothesized that creating a more learner-centered environment by separating students from other live-style conference participants using teleconferencing can provide students more opportunities to speak up freely and engage in self-explanation by improving their psychological safety.

In this study, we measured the impact of a learner-centered approach teleconference on improving students' psychological safety and increasing their self-explanation. This allows experts to understand their clinical reasoning skills and provide more direct, effective, feedback. We also measured whether we could create a more autonomous environment using teleconferencing [17], which has the potential to support their academic achievement [17, 18]. Lastly, we measured students' satisfaction and their conference preferences to explore their perceptions.

## Materials and methods

### Ethics statement

This study was approved by the Ethics Committee of Chiba University School of Medicine (Chiba, Japan). A detailed explanation of the study was given to all participants, who confirmed that they fully understood the information before voluntarily giving informed consent to participate.

### Study design

The study was designed as a quasi-randomized crossover study comparing the effects of two clinical reasoning case conference methods on medical students. The allocation sequence was concealed from all participants enrolled in the study, including medical students, facilitators, fellows, primary care doctors, and professors.

### Study participants

The study was conducted at a single center at Chiba University Hospital. The participants included medical students participating in a two-week general medicine clinical clerkship rotation from September 2019 to February 2020. They rotated as a group of 4–6 students and were able to experience both types of conferences (learner-centered approach teleconference and traditional, live-style), which were held weekly. Each group was assigned through simple randomization using Microsoft Excel 2019 to group 1 or group 2. Participants in group 1 received the learner-centered approach teleconference in the first week, and a traditional, live-style conference in the second week; participants assigned to group 2 received the traditional, live-style conference in the first week, and a learner-centered approach teleconference in the second week. Students who were not able to participate in one of the conferences for any reason (sick leave or school events) were excluded from the study. We had to suspend this study due to the COVID-19 pandemic and decided to analyze the gathered data.

### Procedure

**Traditional live-style conference (control).** Typically, 120-minute case conferences are organized once a week at our institution, with participants consisting of the professor, 15 fellows, five primary care doctors from other facilities, and 4–6 medical students in clinical clerkship (Fig 1). The content of the case presentation unfolds gradually, starting from age, gender, and chief complaint to more details (serial-cue approach) by questioning the presenter [3]. The goal of the conferences was to teach students the clinical reasoning process, including data acquisition, problem representation, hypothesis generation, and illness script selection [19]. The department professor functioned as a moderator and added some educational topics at any time. Typically, a case presentation was prepared by a third- or fourth-year postgraduate student (PGY-3 or PGY-4) from our department. The cases were chosen by department staff from the perspective of clinical reasoning education. All instructions were standardized (S1 Fig) and held just before the conference, and students were instructed that the objective of the conference was to teach clinical reasoning to novices, and their role was to actively participate in the conference. They were assured that they would not be called on or criticized during the conference. When students asked or answered any questions, they had a chance to self-explain their thinking process and were provided feedback by the professor. Students were allowed to share their clinical reasoning process or use the Internet to search for necessary information during this conference.

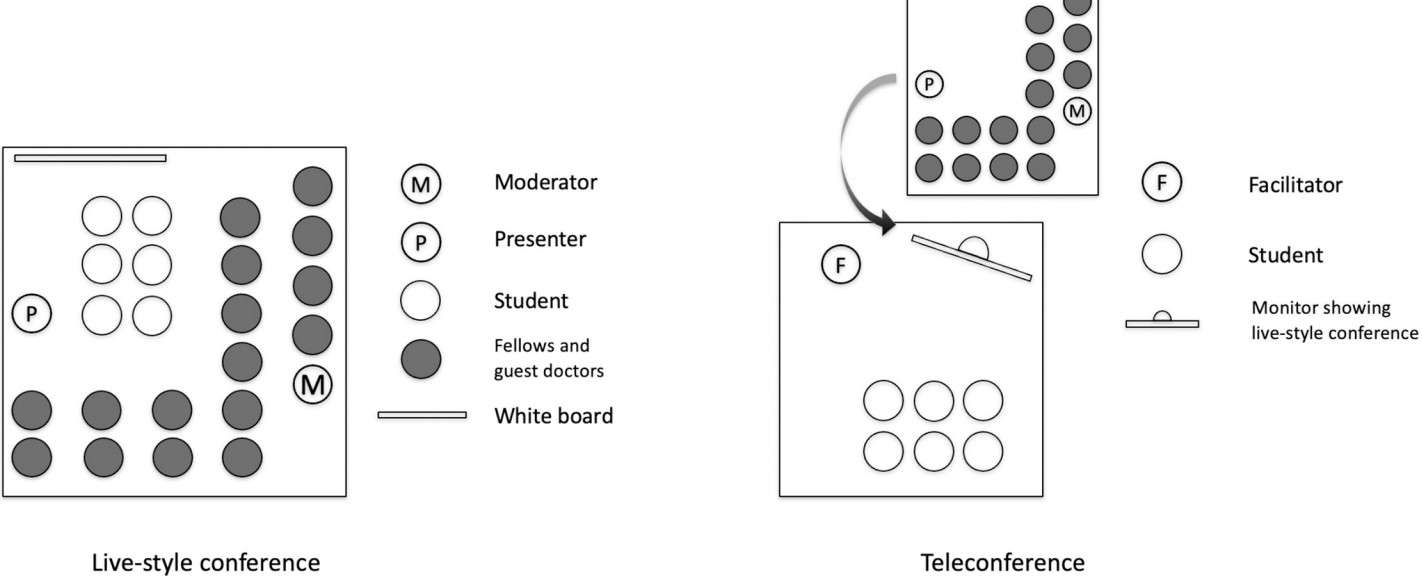

**Fig 1. Traditional live-style conference and learner-centered approach teleconference.**

**Learner-centered approach teleconference (intervention).** On the teleconference day, we created two different rooms. A traditional, live-style conference was organized in one room, and the learner-centered approach teleconference was held in a separate room, with a screen showing the real-time traditional, live-style conference. The learner-centered teleconference consisted of one facilitator and 4–6 medical students in clinical clerkship (Fig 1). The facilitator was randomly chosen from five clinical fellows, PGY-5 or higher (median PGY-7 [5–17]), all males (S1 Table), with sufficient experience to explain the clinical reasoning process to students. Facilitators were allowed to help medical students understand the clinical discussion when it went beyond their level of medical expertise, assist students in solving problems, and answer students' questions in a "learner-centered," question-based way [20, 21]. The facilitator was tutored on their role before each conference, using the instructions established through several focus group discussions (S1 Fig). All instructions were standardized and held just before the conference, and students were similarly instructed that the objective of the teleconference was to teach clinical reasoning to novices, and their role was to actively participate in the conference actively. They were also assured that they would not be called on or criticized during the conference. When students asked or answered any questions, they had a chance to self-explain their thinking process and were provided feedback by the facilitator. During the conference, students were also allowed to share their clinical reasoning process or use the Internet to search for necessary information. To conduct the teleconference, we adopted an online conference system (IPELA$^®$ PCS-XG100S; SONY, Tokyo, Japan) [22].

## Data collection

An outline of the study design is presented in Fig 2. The main assessments made after each conference type were participants' psychological safety, and the number of students who provided self-explanation during the conference. Psychological safety was measured using Edmondson's Psychological Safety Scale [8, 11], consisting of seven items (S3 and S4 Figs). The reliability, validity, and factor structure of the measures have been established [8, 11]. We

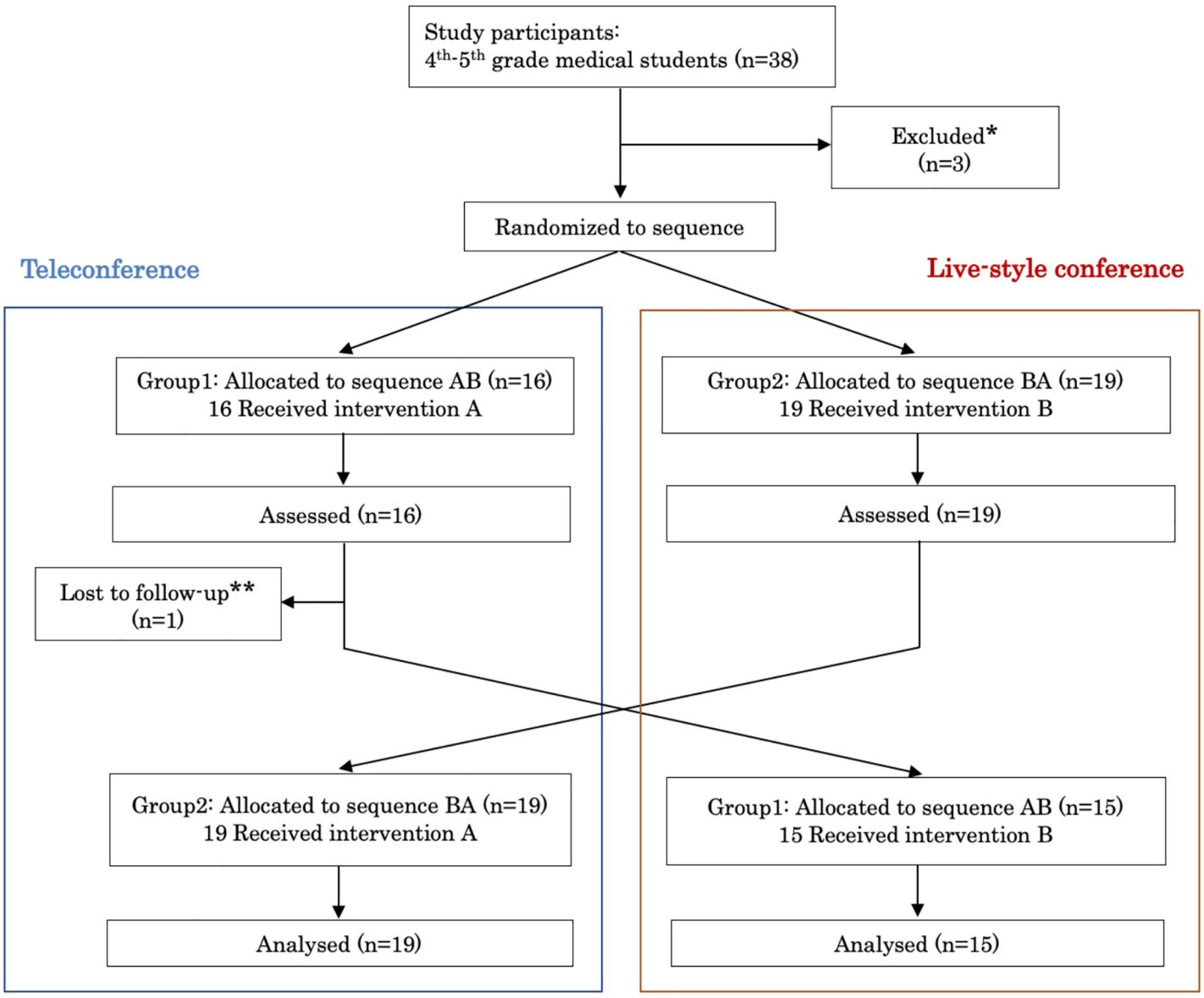

A: Teleconference (Intervention).
B: Live style conference (Control).
\* Students were not able to participate the first conference because they had to do training at an affiliated hospital.
\*\* Student was not able to participate the second conference because of sick leave.

**Fig 2. Study design.**

also analyzed psychological safety for group 1 and group 2 separately. The number of students who provided self-explanations was counted in all conferences. Students' comments that went beyond the information given—specifically, a conjecture of new knowledge about the conference topic—were considered self-explanation [4, 5]. We also measured students' autonomy, satisfaction, and preferences in the two conference types. Autonomy and satisfaction were measured using a 7-point Likert scale, which YH and KS ran focus discussions based on, prior to the published research, to develop the questionnaire from scratch, ranging from 1 "strongly

**Table 1. Participant characteristics.**

| | Total (n = 34) | Group 1 (n = 15) | Group 2 (n = 19) | p-value |
|---|---|---|---|---|
| **Male** | 25 (74%) | 11 (73%) | 14 (74%) | 0.71 |
| **Age, mean (range)** | 23 (21–28) | 23 (21–28) | 23 (21–27) | 0.52 |

disagree" to 7 "strongly agree" (S3 and S4 Figs). For conference preference, students were asked to choose one answer from "Traditional live style conference," "learner-centered approach teleconference," or "Neutral."

This study followed the Consolidated Standards of Reporting Trials (CONSORT) reporting guidelines, as indicated in the flow diagram in S2 Fig.

## Statistical analysis

Descriptive and bivariate analyses were performed to describe and compare participants' psychological safety, autonomy, and satisfaction using students' t-test. The number of students who provided self-explanation was analyzed using the chi-squared test. Gender and age were analyzed using Fisher's t-test and the Mann-Whitney U-test, respectively. All statistical analyses were performed using IBM SPSS version 26.0 (IBM Corp. Armonk, NY), with the level of significance set at $p < 0.05$.

## Results

Thirty-eight medical students (seven groups) participated in the general medicine clinical clerkship rotation during the study period, and 34 students completed the study (Fig 2). Four students were excluded due to absence from their clinical clerkship rotation (three students could not participate in the first conference, and one was not able to participate in the second). The median age of the participants was 23 (21–28) years, and 25 students (71.4%) were male. There were no significant differences in gender or age between the groups (Table 1). Two facilitators participated in the learner-centered approach teleconference twice, and three facilitators participated once (S1 Table).

Six out of seven psychological safety items scored higher in the learner-centered approach teleconference than in the traditional, live-style conference (Table 2). The sub-analysis of each conference showed that three out of seven items in a traditional, live-style conference (control)

**Table 2. Edmondson's Psychological Safety Scale.**

| Items | Intervention mean, SD (n = 34) | Control mean, SD (n = 34) | p-value |
|---|---|---|---|
| 1. If you make a mistake on this team, it is often held against you. (R) | 2.0 ± 1.2 | 2.8 ± 1.4 | <0.01** |
| 2. Members of this team are able to bring up problems and tough issues. | 4.7 ± 1.2 | 5.1 ± 1.1 | <0.01** |
| 3. Members of this team sometimes reject others for being different. (R) | 1.9 ± 1.0 | 2.9 ± 1.5 | <0.01** |
| 4. It is safe to take a risk on this team. | 5.1 ± 1.4 | 4.3 ± 1.7 | <0.01** |
| 5. It is difficult to ask other members of this team for help. (R) | 2.4 ± 1.4 | 3.2 ± 1.7 | <0.01** |
| 6. No one on this team would deliberately act in a way that undermines my efforts. | 5.9 ± 1.4 | 5.1 ± 1.8 | <0.01** |
| 7. Working with members of this team, my unique skills and talents are valued and utilized. | 4.5 ± 1.4 | 3.9 ± 1.8 | <0.01** |

*$p<0.05$

**$p<0.01$

Intervention: Learner-centered approach teleconference, Control: Traditional live style conference.

Note: Each item was measured on a 7-point Likert scale: 1 = very inaccurate; 7 = very accurate.

(R): reversed score.

**Table 3. Edmondson's Psychological Safety Scale (control).**

| Items | Group1 mean, SD (n = 15) | Group2 mean, SD (n = 19) | p-value |
|---|---|---|---|
| 1. If you make a mistake on this team, it is often held against you. (R) | 2.7 ± 1.5 | 2.8 ± 1.4 | 0.91 |
| 2. Members of this team are able to bring up problems and tough issues. | 5.5 ± 0.9 | 4.8 ± 1.2 | 0.08 |
| 3. Members of this team sometimes reject others for being different. (R) | 2.3 ± 1.0 | 3.5 ± 1.6 | 0.02* |
| 4. It is safe to take a risk on this team. | 4.7 ± 1.8 | 4.1 ± 1.6 | 0.31 |
| 5. It is difficult to ask other members of this team for help. (R) | 2.7 ± 1.7 | 3.6 ± 1.7 | 0.12 |
| 6. No one on this team would deliberately act in a way that undermines my efforts. | 4.3 ± 2.1 | 5.7 ± 1.3 | 0.03* |
| 7. Working with members of this team, my unique skills and talents are valued and utilized. | 4.7 ± 1.9 | 3.3 ± 1.5 | 0.02* |

*p<0.05

**p<0.01

Intervention: Learner-centered approach teleconference, Control: Traditional live style conference.

Note: Each item was measured on a 7-point Likert scale: 1 = very inaccurate; 7 = very accurate.

(R): reversed score.

were significantly different (Table 3). No items in the learner-centered approach teleconference (intervention) showed a significant difference (Table 4). The autonomy score was higher in the learner-centered approach teleconference (intervention style 4.4 ± 1.5 vs. control style 3.0 ± 2.0, $p$<0.01) (Table 5). There was no significant difference in satisfaction between conferences, with a high score for both types (intervention style 5.5 ± 1.2 vs. control style 5.4 ± 1.1, $p$ = 0.66) (Table 5). The teleconference was preferred by 15 students (44.1%), and the traditional conference by 8 students (23.5%), while 11 students were neutral (32.4%). Twenty-nine

**Table 4. Edmondson's Psychological Safety Scale (intervention).**

| Items | Group1 mean, SD (n = 15) | Group2 mean, SD (n = 19) | p-value |
|---|---|---|---|
| 1. If you make a mistake on this team, it is often held against you. (R) | 1.7 ± 1.2 | 2.3 ± 1.1 | 0.19 |
| 2. Members of this team are able to bring up problems and tough issues. | 4.7 ± 1.3 | 4.6 ± 1.2 | 0.82 |
| 3. Members of this team sometimes reject others for being different. (R) | 1.5 ± 0.5 | 2.1 ± 1.2 | 0.11 |
| 4. It is safe to take a risk on this team. | 5.3 ± 1.3 | 4.9 ± 1.4 | 0.37 |
| 5. It is difficult to ask other members of this team for help. (R) | 2.1 ± 1.4 | 2.7 ± 1.4 | 0.26 |
| 6. No one on this team would deliberately act in a way that undermines my efforts. | 6.3 ± 1.5 | 5.6 ± 1.3 | 0.21 |
| 7. Working with members of this team, my unique skills and talents are valued and utilized. | 4.8 ± 1.4 | 4.4 ± 1.4 | 0.44 |

*p<0.05

**p<0.01

Intervention: Learner-centered approach teleconference, Control: Traditional live style conference.

Note: Each item was measured on a 7-point Likert scale: 1 = very inaccurate; 7 = very accurate.

(R): reversed score.

**Table 5. Autonomy and satisfaction for both conferences.**

| Secondary outcome | Intervention mean, SD (n = 34) | Control mean, SD (n = 34) | p-value |
|---|---|---|---|
| **Autonomy** | 4.4 ± 1.5 | 3.0 ± 2.0 | <0.01** |
| **Satisfaction** | 5.5 ± 1.2 | 5.4 ± 1.1 | 0.66 |

*p<0.05

**p<0.01

Intervention: Learner-centered approach teleconference, Control: Traditional live style conference.

Note: Autonomy and Satisfaction were measured on a 7-point Likert scale: 1 = strongly disagree; 7 = strongly agree.

(85.3%) students provided self-explanation at least once in the learner-centered teleconference compared to ten students (29.4%) in the traditional live conference (p<0.01).

## Discussion

This study confirmed that a learner-centered approach teleconference can create a psychologically safe environment where learners can speak up more easily, and as such, this can be a new method to successfully teach clinical reasoning to medical students.

This study investigated the impact of the learner-centered approach teleconference on students' psychological safety and self-explanation. The study considered psychological safety to be "a shared belief held by members of a team that the team is safe for interpersonal risk taking," [11] which affects interpersonally risky behaviors [8, 10]. An absence of psychological safety, which is often felt at a lower status within a professional hierarchy—such as a conference with department staff members—can prevent students from speaking up during the conference, even when they know they have something to contribute [6, 11]. This adversely affects students' autonomy in actively participating in the conference and makes it difficult for department staff to assess students' clinical reasoning process and provide them direct and effective feedback. We believe that improving students' psychological safety is key to promoting clinical reasoning education during lively case conferences, and we have thus tried to create a more psychologically safe environment.

Learner-centered approach using small groups with a tutor, such as in PBL, involves the learner as an active participant [14]. We hypothesized that combining a small group, a learner-centered approach with the conventional lively case conference using teleconferencing can help students feel safe and speak up. Our main goal was to diminish the professional-derived status hierarchy by introducing a more learner-centered environment with only students, and one facilitator. The facilitator was assigned to help students guide the discussion in the right direction, assisting in the lack of knowledge, and giving effective feedback in a "learner-centered," question-based way [20, 21].

The results showed that six out of seven items assessing psychological safety significantly improved in the learner-centered teleconference. In addition, more students in the learner-centered approach teleconference could speak up during the conference (29 students vs. 10 students, P<0.01), consistent with previous studies [8, 10]. These results enable educators to provide students with more direct, effective feedback during conferences. Only one item, "Members of this team are able to bring up problems and tough issues" showed the opposite result. This may be due to the difficulty of asking questions to the presenter in a remote environment. Indeed, students were not able to ask questions to the presenter directly because the microphone connecting the separate rooms was one-way from the live conference room to the teleconference room. System changes that allow teleconference participants to interact with live-style conferences can potentially overcome this problem; however, this change also risks

the psychological safety of novices. Another possibility is that students may find it easier to discuss problems and tough issues during the traditional live-style conference when they have department members around them, compared to a teleconference, because department members can ask questions to the presenter instead of students. Therefore, further studies are required.

In this study, we also measured students' autonomy, satisfaction, and conference preferences to identify learners' perceptions. While we used an original 7-point Likert scale, the autonomy score was higher in the teleconference, consistent with a previous report [17]. This result was expected, as we anticipated that a learner-centered conference would augment students' relatedness and engagement, potentially fostering autonomy. In fact, some studies reported that small-group teaching supports autonomy in medical education, enhances autonomous motivation for medical study, and makes students autonomy-supportive in their future medical practice [23].

Although there was no significant difference in each group's satisfaction scores, which were high for both conferences, more students chose teleconference as their preference, which was opposite to previous reports [24, 25]. Before starting this study, we expected that students would prefer to participate in traditional face-to-face live-style conferences because students can receive direct feedback from the professor and other department staff. Presumably, the learner-centered style, which enables novice learners to discuss easily, has contributed to high satisfaction. In addition, students in the teleconference could receive direct feedback from experienced facilitators, which may contribute to their high satisfaction and preference. Although the results should be accepted cautiously as this is a single study with relatively few participants, we believe that this learner-centered approach has the potential to effectively teach clinical reasoning by providing learners with more opportunities to speak up during the conference. Furthermore, this conference style can be organized at many facilities simultaneously and would be a practical teaching method even during the COVID-19 pandemic era, where maintaining social distance is essential.

This study has some limitations. First, as mentioned above, this was a single-site study conducted in Japan, with relatively few participants. However, our facility is a national university that represents a standard medical school in Japan. Although the study was interrupted due to the COVID-19 crisis, we believe it can support the introduction of a small-group, learner-centered approach teleconference that can potentially be conducted safely in this new era of social distancing. Passive attitudes, such as the tendency to avoid failure caused by cultural differences, may impact the results. However, we assume that the results can encourage students who have difficulty speaking up during the conference due to low psychological safety globally. Second, there is a possibility that the facilitator's skills may directly affect the effectiveness of the conference. We chose five experienced facilitators, applied randomly to each case, using a standardized facilitator guide (S1 Fig) to minimize the influence. In addition, we considered that the difficulty of the case could affect students' psychological safety. Each case was chosen randomly by faculty members to teach clinical reasoning. Cases that were too complicated or straightforward were not included. Third, Edmondson's Psychological Safety Scale was translated into Japanese to help students understand the questions clearly and precisely (S3 Fig). Furthermore, autonomy and satisfaction were measured using an original 7-point Likert scale with no validation, and further studies may be needed. Fourth, it is difficult to apply this result to students who are already inclined to speak up in front of others. Nevertheless, our study aimed to help students with low psychological safety and difficulty speak up during conferences. Fifth, it was difficult to remove confounding factors between the professor and the facilitator. It was assumed that the existence of the professor during the conference could negatively affect students' psychological safety and participation in the conference. Professors

and facilitators were not allowed to call on any student during the study. Moreover, before every conference, students were informed that they would not be called on and were free to discuss. Finally, we considered the bias to carry-over the decision to assess psychological safety or self-explanation. We conducted a sub-analysis of the results for groups 1 and 2 separately, as shown in Tables 3 and 4. Three items in a traditional live-style conference (control) in Table 3 showed the possibility of some residual effects, and no items in the learner-centered approach teleconference (intervention) showed significant differences. Among the three items that showed a significant difference, two (questions 3 and 7) demonstrated that group 1 had better psychological safety than group 2. We believe that conducting a traditional, live-style conference in the second week can improve learners' psychological safety because they know what to expect from it. Moreover, students might know each other better in the second week of a conference, which helps facilitate teamwork. Although there were no significant differences, the $2^{nd}$ and $5^{th}$ items demonstrated a trend toward the traditional, live-style conference in the second week having higher psychological safety than in the first week. However, the $6^{th}$ item does not support this trend, and additional studies are required to assess this reasoning. Since this was a single event, we expect the possibility of residual effects to be low.

This study has shown the possibility that a learner-centered approach teleconference can successfully teach clinical reasoning to medical students and is feasible even in the COVID-19 pandemic era. However, further studies are required to provide more comprehensive evidence.

## Conclusions

A learner-centered approach teleconference can create a psychologically safe environment for novice learners. This environment could also help learners speak more comfortably.

## Supporting information

**S1 Table. Facilitator characteristics.**
(PDF)

**S2 Table. Data (in Japanese/English).**
(XLSX)

**S1 Fig. Facilitator guide.**
(PDF)

**S2 Fig. CONSORT.**
(PDF)

**S3 Fig. Questionnaire (Japanese).** Questionnaire used in the study (in Japanese).
(PDF)

**S4 Fig. Questionnaire (English).** Questionnaire used in the study (in English).
(PDF)

## Acknowledgments

We would like to thank Editage (www.editage.com) for English language editing.

## Author Contributions

**Conceptualization:** Yoji Hoshina, Kiyoshi Shikino, Yosuke Yamauchi, Daiki Yokokawa, Tomoko Tsukamoto, Kazutaka Noda, Masatomi Ikusaka.

**Data curation:** Yoji Hoshina, Daiki Yokokawa.

**Formal analysis:** Yoji Hoshina, Kiyoshi Shikino.

**Investigation:** Yoji Hoshina, Kiyoshi Shikino, Yosuke Yamauchi, Yasutaka Yanagita, Daiki Yokokawa.

**Methodology:** Yoji Hoshina, Kiyoshi Shikino, Yosuke Yamauchi, Yasutaka Yanagita, Tomoko Tsukamoto, Kazutaka Noda, Masatomi Ikusaka.

**Project administration:** Yoji Hoshina, Masatomi Ikusaka.

**Supervision:** Kiyoshi Shikino, Masatomi Ikusaka.

**Validation:** Kiyoshi Shikino.

**Writing – original draft:** Yoji Hoshina.

**Writing – review & editing:** Kiyoshi Shikino, Yosuke Yamauchi, Yasutaka Yanagita, Daiki Yokokawa, Tomoko Tsukamoto, Kazutaka Noda, Takanori Uehara, Masatomi Ikusaka.

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
