## [Decision Letter · Decision Letter 0]

16 Mar 2021

PONE-D-21-05048

Does the learner-centered approach using teleconference improve medical students’ psychological safety and self-explanation in clinical reasoning conferences? A crossover study

PLOS ONE

Dear Dr. Kiyoshi,

Thank you for submitting your manuscript to PLOS ONE. After careful consideration, we feel that it has merit but does not fully meet PLOS ONE’s publication criteria as it currently stands. Therefore, we invite you to submit a revised version of the manuscript that addresses the points raised during the review process.

We look forward to receiving your revised manuscript.

Kind regards,

Gwo-Jen Hwang

Academic Editor

PLOS ONE

Journal Requirements:

2. Thank you for stating the following financial disclosure: 'No'

Reviewers' comments:

Reviewer's Responses to Questions

**Comments to the Author**

1. Is the manuscript technically sound, and do the data support the conclusions?

Reviewer #1: Partly

Reviewer #2: No

2. Has the statistical analysis been performed appropriately and rigorously? 

Reviewer #1: Yes

Reviewer #2: Yes

3. Have the authors made all data underlying the findings in their manuscript fully available?

Reviewer #1: No

Reviewer #2: No

4. Is the manuscript presented in an intelligible fashion and written in standard English?

Reviewer #1: No

Reviewer #2: Yes

5. Review Comments to the Author

Reviewer #1: This is an interesting paper on a timely subject: the use of teleconferencing for medical student clinical reasoning conferences. I have several suggestions for improving the paper.

First, the paper needs careful attention to grammar and diction. Although much of the paper is well-written, there are many cases where improvement is needed. For example, line 83 ends prematurely. In line 85, the word "majored" should be replaced with "measured". Line 125-126 "they won't be call on" should be "they won't be called on" (this is redundant with line 145).

Second, the term "psychological safety" should be defined.

Third, the results for Group 1 and Group 2 should be reported separately. I am curious about whether having the intervention first altered the experience Group 1 of the subsequent control session, as compared to the control session of Group 2.

Fourth, the gender of the faculty should be reported. Were there different results when the gender of the faculty matched tihe students?

Fifth, Figure 3 does not add to the paper over what is in the text (lines 191-193).

Reviewer #2: Introduction section, for more contribution, it would be better definition the“teleconference”, why it's important for the impact of the learner-centered approach.

2.Introduction Section, authors should address more Psychological safety-related “teleconference” references.for example, why authors want to explore Psychological Safety use “teleconference”.

3.Study participants section, it would better to address how to divided Participants into groups 1 and group 2.

4.Page 8, line 133-151, Learner-centered approach teleconference (Intervention), for readable, suggest authors rewording, what kind of Learner-centered approach teleconference, the teleconference content includes what kind of professional knowledge?

5.Page 9, Data collection section, what the “Edmondson's psychological safety scale” and autonomy, satisfaction, and preference scale Reliability , it woud better cite reference.

6.research framework and methodology should address more:Why authors want to measure psychological safety, autonomy, satisfaction and preference.

7.The discussion and Conclusions section should follow the research aims.

6. PLOS authors have the option to publish the peer review history of their article (what does this mean?). If published, this will include your full peer review and any attached files.

Reviewer #1: No

Reviewer #2: No

---

## [Author Response · Author response to Decision Letter 0]

25 Apr 2021

April 23, 2021

Dear Editors and Reviewers

Title: Does a learner-centered approach using teleconference improve medical students’ psychological safety and self-explanation in clinical reasoning conferences? A crossover study

Reference number: PONE-D-21-05048

Dear Editors and Reviewers,

Thank you for your e-mail on March 17, 2021, regarding our manuscript, ” Does a learner-centered approach using teleconference improve medical students’ psychological safety and self-explanation in clinical reasoning conferences? A crossover study,” and for the valuable comments of the associated editor and referees. I have attached our marked-up copy of our manuscript, revised manuscript, as well as a point-by-point response to the associated editor and referees’ comments.

Responses to the comments by the academic editor and reviewers.

Responses to the comments by the reviewer 1:

Q1. The paper needs careful attention to grammar and diction. Although much of the paper is well-written, there are many cases where improvement is needed. For example, line 83 ends prematurely. In line 85, the word "majored" should be replaced with "measured". Line 125-126 "they won't be call on" should be "they won't be called on" (this is redundant with line 145).

Reply:

We sincerely appreciate all your important comments. We paid careful attention to grammar and diction and submitted it to Editage (www.editage.com) for proofreading. All changes are highlighted in red on the marked-up copy of our manuscript.

Q2. The term "psychological safety" should be defined.

Reply:

We appreciate your suggestion regarding the use of “psychological safety.” As suggested, we added sentences with citation that define the term “psychological safety” in the introduction section on lines 75-83 (Page 4-5).

Change:

Lines 75-83 (Page 4-5).

“Psychological safety, the degree to which people view the environment as conducive to interpersonally risky behaviors, like speaking up or asking for help, impacts the degree to which individual or group learning can occur. In fact, a profession-derived status hierarchy has a positive relationship with psychological safety, and individuals who experience greater psychological safety are more likely to speak up and present new ideas. A degree of team psychological safety, which is a shared belief that the team is safe for interpersonal risk-taking, is key to facilitate learning behavior by alleviating excessive concern about others’ reactions to actions that have the potential for embarrassment or threat.”

Q3. The results for Group 1 and Group 2 should be reported separately. I am curious about whether having the intervention first altered the experience Group 1 of the subsequent control session, as compared to the control session of Group 2.

Reply:

Thank you very much for your excellent suggestion. We analyzed “psychological safety” results for Group1 and Group2 separately and added Table 2 (Group1) and Table 3 (Group2). The previous Table 2 result is now in Table 4.

Although there was no significant difference in Group 2, three items in Group 1 showed significant differences, which can be considered as a residual effect. We reported the result in the limitation section.

Changes:

We added Table 2 and Table 3. The previous Table 2 result is now in Table 4 (Page 14-15).

We also reported the result in lines 219-222 (Page 13).

“The sub-analysis of each conference showed that three out of seven items in a traditional, live-style conference (control) were significantly different (Table 2). No items in the learner-centered approach teleconference (intervention) showed significant difference (Table 3).”

Additionally, we added some discussion on lines 345-358 (Page 21-22). 

“We conducted a sub-analysis of the results for groups 1 and 2 separately, shown in Tables 2 and 3. Three items in a traditional live-style conference (control) in Table 2 showed the possibility of some residual effects, and no items in the learner-centered approach teleconference (intervention) showed significant differences. Among the three items that showed a significant difference, two (questions 3 and 7) demonstrated that group 1 had better psychological safety than group 2. We consider that conducting a traditional, live-style conference in the second week can improve learners’ psychological safety because they know what to expect from it. Moreover, students might know each other better in the second week of a conference, which helps facilitate teamwork. Although there were no significant differences, the 2nd and 5th items demonstrated a trend towards the traditional, live-style conference in the second week having higher psychological safety than in the first week. However, the 6th item does not support this trend and additional studies are required to assess the reasoning for this. Since this was a one-time, single event, we expect the possibility of residual effects to be low.

Q4. The gender of the faculty should be reported. Were there different results when the gender of the faculty matched the students?

Reply:

We apologize for our insufficient explanation regarding the gender of the faculty. We added the information in the method section (Learner-centered approach teleconference). All faculty members who were assigned as a facilitator during this study were unintentionally males.

Change:

Line 166 (Page 10).

“The facilitator was randomly chosen from five clinical fellows, PGY-5 or higher (median PGY-7 (5-17)), all males, with sufficient experience to explain the clinical reasoning process to students.”

Q5. Figure 3 does not add to the paper over what is in the text (lines 191-193).

Reply:

Thank you for your comment. We decided to delate Figure 3 as indicated.

Change:

We delated Figure 3.

Responses to the comments by the reviewer 2:

Q1. Introduction section, for more contribution, it would be better definition the “teleconference”, why it's important for the impact of the learner-centered approach.

Reply:

We apologize for our insufficient explanation regarding the use of “teleconference.” We added the definition of the word “teleconference” in the introduction section with a citation on line 93-96 (Page 5-6).

Change:

Lines 93-96 (Page 5-6). 

“Teleconferences, which describe the creation of two or more learning environments where users can engage in real-time communication to exchange presentations or data via audio or live video, can constitute a learner-centered model by helping students learn on their own, as well as in collaboration with others.”

Q2. Introduction Section, authors should address more Psychological safety-related “teleconference” references.for example, why authors want to explore Psychological Safety use “teleconference”.

Reply: 

Thank you very much for your important comments. Although we could not find any previous studies that measured psychological safety during teleconferences, we hypothesized that creating a more learner-centered environment by separating students from other live style conference participants by using teleconference can create a psychologically safe environment to give students more opportunity to speak up freely and engage in self-explanation.

Change:

Lines 99-104 (Page 6).

“Although we could not find any previous studies that measured psychological safety during teleconferences, we hypothesized that creating a more learner-centered environment by separating students from other live-style conference participants using teleconference can give students more opportunities to speak up freely and engage in self-explanation by improving their psychological safety.”

Q3. Study participants section, it would be better to address how to divide Participants into groups 1 and group 2.

Reply:

Thank you very much for your excellent suggestion. Each group was randomly assigned through simple randomization using Microsoft Excel 2019.

Change:

Lines 125-126 (Page 7).

“Each group was randomly assigned through simple randomization, using Microsoft Excel 2019, to group 1 or group 2.”

Q4. Page 8, line 133-151, Learner-centered approach teleconference (Intervention), for readable, suggest authors rewording, what kind of Learner-centered approach teleconference, the teleconference content includes what kind of professional knowledge?

Reply:

Thank you very much for your important comments. We decided to add more explanation to the learner-centered approach teleconference in lines 160-163.

Further, we added the objectives of conferences to explain the professional knowledge included in each conference. 

Changes: 

Lines 160-163 (Page 9)

“On the teleconference day, we created two different rooms. A traditional, live-style conference was organized in one room, and the learner-centered approach teleconferences were held in a separate room with a screen showing the real-time traditional, live-style conference.”

Lines 142-144 (Page 8).

“The goal of the conferences was to teach students the clinical reasoning process, including data acquisition, problem representation, hypothesis generation, and illness script selection”

Q5. Page 9, Data collection section, what the “Edmondson's psychological safety scale” and autonomy, satisfaction, and preference scale Reliability , it woud better cite reference.

Reply:

Thank you for your comment. We cited the reference for Edmondson’s psychological safety in line 186 (Page 11). 

Autonomy and satisfaction were scaled using a 7-point Likert scale, which YH and KS ran focus discussions based on, prior to the published research, to develop the questionnaire form from scratch, ranging from 1 “strongly disagree” to 7 “strongly agree.” For conference preference, students were asked to choose one answer from “Traditional live style conference,” “learner-centered approach teleconference,” or “Neutral.”

Changes:

Lines 185-198 (Pages 11-12).

“Psychological safety was measured using Edmondson's psychological safety scale [8, 11], consisting of seven items (S3, S4 Fig). The reliability, validity, and factor structure of the measure have been established [8, 11]. We also analyzed psychological safety for group 1 and group 2 separately. The number of students who provided self-explanation was counted during all conferences. Students’ comments that went beyond the information given—specifically, a conjecture of new knowledge about the conference topic—were considered self-explanation [4, 5]. We also measured the autonomy, satisfaction, and preference of the two conferences. Autonomy and satisfaction were scaled using a 7-point Likert scale, which YH and KS ran focus discussions based on, prior to the published research, to develop the questionnaire form from scratch, ranging from 1 “strongly disagree” to 7 “strongly agree” (S3, S4 Fig). For conference preference, students were asked to choose one answer from “Traditional live style conference,” “learner-centered approach teleconference,” or “Neutral.”

Q6. Research framework and methodology should address more: Why authors want to measure psychological safety, autonomy, satisfaction and preference.

Reply:

We sincerely appreciate your comment. We wanted to measure psychological safety to compare the two types of conferences and quantify if we could create a psychologically safer environment where students could speak up more easily. We also measured whether we could create a more autonomous environment using teleconference, which has the potential to support their academic achievement. Lastly, we measured students’ satisfaction and their conference preference to explore their perception.”

Changes:

Lines 104-110 (Page 6)

“In this study, we measured the impact of a learner-centered approach teleconference on improving students’ psychological safety and increasing self-explanation. This allows experts to understand their clinical reasoning skills and provide more effective and efficient feedback. We also measured whether we could create a more autonomous environment using teleconference [17], which has the potential to support their academic achievement [17, 18]. Lastly, we measured students’ satisfaction and their conference preference to explore their perception.”

Q7. The discussion and Conclusions section should follow the research aims.

Reply:

Thank you very much for your invaluable comments. We changed the discussion and conclusions so that they follow the research aims.

We have also added Data availability on the “Supporting information.”

For “Financial disclosure,” the authors received no specific funding for this work.

We hope that the revised manuscript contains suitable responses to the comments, and we think that it has been significantly improved over the initial submission. We trust that our manuscript is now suitable for publication in PLOS ONE.

Thank you in advance for your kind consideration of our work.

Sincerely yours,

Kiyoshi Shikino, MD, PhD, MHPE　

Department of General Medicine, Chiba University Hospital

1-8-1, Inohana, Chuo-ku, Chiba-city, Chiba, Japan

Tel. +81-43-222-7171 (Ext. 6438); +81-43-224-4758 (Direct line)

Fax. +81-43-224-4758

E-Mail: kshikino@gmail.com

---

## [Decision Letter · Decision Letter 1]

17 May 2021

PONE-D-21-05048R1

Does the learner-centered approach using teleconference improve medical students’ psychological safety and self-explanation in clinical reasoning conferences? A crossover study

PLOS ONE

Dear Dr. Kiyoshi,

Thank you for submitting your manuscript to PLOS ONE. After careful consideration, we feel that it has merit but does not fully meet PLOS ONE’s publication criteria as it currently stands. Therefore, we invite you to submit a revised version of the manuscript that addresses the points raised during the review process.

We look forward to receiving your revised manuscript.

Kind regards,

Gwo-Jen Hwang

Academic Editor

PLOS ONE

Journal Requirements:

Reviewers' comments:

Reviewer's Responses to Questions

**Comments to the Author**

1. If the authors have adequately addressed your comments raised in a previous round of review and you feel that this manuscript is now acceptable for publication, you may indicate that here to bypass the “Comments to the Author” section, enter your conflict of interest statement in the “Confidential to Editor” section, and submit your "Accept" recommendation.

Reviewer #1: All comments have been addressed

Reviewer #2: (No Response)

2. Is the manuscript technically sound, and do the data support the conclusions?

Reviewer #1: Yes

Reviewer #2: No

3. Has the statistical analysis been performed appropriately and rigorously? 

Reviewer #1: Yes

Reviewer #2: No

4. Have the authors made all data underlying the findings in their manuscript fully available?

Reviewer #1: Yes

Reviewer #2: No

5. Is the manuscript presented in an intelligible fashion and written in standard English?

Reviewer #1: Yes

Reviewer #2: No

6. Review Comments to the Author

Reviewer #1: (No Response)

Reviewer #2: Dear Authors,

1. The abstract section, should be rewording depend on your results, Figure3 and Figure 2 not match, please check.

2. Page 11, line181-185, Results section, “…after the first conferences. The median age of the participants were 23 (21–28) years, and 25 students 184 (71.4%) were men. Two facilitators participated in learner-centered approach teleconference twice and 3 facilitators participated once...” for readable and contribution, suggest authors add descriptive statistics chart.

3. For easy read, suggest authors add participants sample size in the Table 1. and Table 2.

4. Page 11, Results section, should be rewording too, because “learner192 centered approach teleconference was preferred by fifteen students (44.1%), traditional 193 conferences by eight students (23.5%), and neutral by eleven students (32.4%) (Fig 3). 194 Twenty-nine (85.3%) students did self-explanation at least once in the learner-centered195 approach teleconference compared to ten students (29.4%) in the traditional live 196 conference style (p<0.01).” but authors’ Figure 3 presented that teleconference (44.1%) and N=15 not match Figure 2. teleconference group and N=19.

5. The discussion and conclusions section should follow the Results section.

6.It is suggested that authors have the paper proofread by a professional proofreader who is a native English speaker.

---

## [Author Response · Author response to Decision Letter 1]

2 Jun 2021

May 31, 2021

Dear Editors and Reviewers

Title: Does a learner-centered approach using teleconference improve medical students’ psychological safety and self-explanation in clinical reasoning conferences? A crossover study

Reference number: PONE-D-21-05048

Dear Editors and Reviewers,

Thank you for your e-mail on May 17, 2021, regarding our manuscript, ”Does a learner-centered approach using teleconference improve medical students’ psychological safety and self-explanation in clinical reasoning conferences? A crossover study,” and for the valuable comments of the associated editor and referees. I have attached our marked-up copy of our manuscript, revised manuscript, as well as a point-by-point response to the associated editor and referees’ comments.

Responses to the comments by the academic editor and reviewers.

Responses to the comments by the reviewer #2:

Q1. The abstract section, should be rewording depend on your results, Figure3 and Figure 2 not match, please check.

Q4. Page 11, Results section, should be rewording too, because “learner centered approach teleconference was preferred by fifteen students (44.1%), traditional conferences by eight students (23.5%), and neutral by eleven students (32.4%) (Fig 3). Twenty-nine (85.3%) students did self-explanation at least once in the learner-centered approach teleconference compared to ten students (29.4%) in the traditional live conference style (p<0.01).” but authors’ Figure 3 presented that teleconference (44.1%) and N=15 not match Figure 2. teleconference group and N=19.

Reply:

We sincerely appreciate all your important comments. We reworded the abstract section and the result section as described below. First, we deleted line 44 and lines 49-51 on page 3 (on Revised manuscript with track changes, abstract section) to make the abstract concise and to focus more on psychological safety and the number of students who speak up during each conference.

Figure 3 was delated during the previous revision because it was pointed out that it doesn’t add to the paper over the text on lines 228-230 (page 14). N=15 and n=19 on Figure 2 is the number of students assigned to Group 1 and Group 2, respectively. Additionally, we added “n=34” next to the name of each conference in Figure 2 for readable purposes.

Change:

We delated line 44 and lines 49-51 on Page 3 (on Revised manuscript with track changes).

Figure 3 was delated during the previous revision.

“n=34” was added on Figure 2, next to the name of each conference.

Q2. Page 11, line181-185, Results section, “…after the first conferences. The median age of the participants were 23 (21–28) years, and 25 students 184 (71.4%) were men. Two facilitators participated in learner-centered approach teleconference twice and 3 facilitators participated once...” for readable and contribution, suggest authors add descriptive statistics chart.

Reply:

We appreciate your suggestion for adding a descriptive statistics chart. We added a new Table (Table 1) which describes the statistics chart of participants. Additionally, we added a new supplement (S1 Table) which describes the character of facilitators. 

Statistical analysis for gender and age of students was added on line 204-205, Page 12.

Because we added a new Table 1, the previous Table 1 and Table 2 are now Table 2 and Table 3, respectively.

Change:

We added a new Table 1 and S1 Table.

We added line 204-205 (page 12)

 “Gender and age were analyzed using Fisher’s t-test and the Mann-Whitney U-test, respectively.”

Previous Tables 1, 2, 3, and 4 are now Tables 2, 3, 4, and 5, respectively.

Q3. For easy read, suggest authors add participants sample size in the Table 1. and Table 2.

Reply:

Thank you very much for your excellent suggestion. We added sample size in Table 1 and Table 2. Also, we added sample size on new Table 3 and Table 4.

Because we added a new Table 1, which is the participant characteristic, the previous Tables 1, 2, 3, 4 are now Table 2, 3, 4, and 5, respectively.

Changes:

We added sample size in new Table 2, Table 3, Table 4, and Table 5. 

Q5. The discussion and conclusions section should follow the Results section.

Reply:

Thank you very much for your invaluable comments. During the previous revision, we added the subanalysis and had reworded the discussion and conclusion sections based on the results and limitations. In this revision, we have reworded additional changes so the discussion and conclusion follow the result section. 

Change:

Lines 316-318 (Page 21)

Add “In addition, students in the teleconference could receive direct feedback from experienced facilitator, which may contribute to their high satisfaction and preference.”

Line 390-391 (Page 25) on revised manuscript with track changes.

Delated “enabling educators to assess their understanding levels more effectively.” 

Additional changes are shown in red color on revised manuscript with track changes.

Q6. It is suggested that authors have the paper proofread by a professional proofreader who is a native English speaker.

Reply:

We sincerely appreciate all your important comments. We submitted our manuscript to Editage (www.editage.com), which is a native English speaker for proofreading. All changes are highlighted in red on the marked-up copy of our manuscript.

For “Financial disclosure,” the authors received no specific funding for this work.

We hope that the revised manuscript contains suitable responses to the comments, and we think that it has been significantly improved over the initial submission. We trust that our manuscript is now suitable for publication in PLOS ONE.

Thank you in advance for your kind consideration of our work.

Sincerely yours,

Kiyoshi Shikino, MD, PhD, MHPE　

Department of General Medicine, Chiba University Hospital

1-8-1, Inohana, Chuo-ku, Chiba-city, Chiba, Japan

Tel. +81-43-222-7171 (Ext. 6438); +81-43-224-4758 (Direct line)

Fax. +81-43-224-4758

E-Mail: kshikino@gmail.com

---

## [Editor Report · Decision Letter 2]

15 Jun 2021

Does the learner-centered approach using teleconference improve medical students’ psychological safety and self-explanation in clinical reasoning conferences? A crossover study

PONE-D-21-05048R2

Dear Dr. Kiyoshi,

We’re pleased to inform you that your manuscript has been judged scientifically suitable for publication and will be formally accepted for publication once it meets all outstanding technical requirements.

Kind regards,

Gwo-Jen Hwang

Academic Editor

PLOS ONE
---

## [Editor Report · Acceptance letter]

28 Jun 2021

PONE-D-21-05048R2 

Does a learner-centered approach using teleconference improve medical students’ psychological safety and self-explanation in clinical reasoning conferences? A crossover study 

Dear Dr. Shikino:

I'm pleased to inform you that your manuscript has been deemed suitable for publication in PLOS ONE. Congratulations! Your manuscript is now with our production department. 

Kind regards, 

on behalf of

Dr. Gwo-Jen Hwang 

Academic Editor

PLOS ONE